# Study on Shearing Behavior of Circular Concrete-Filled CFRP (Carbon Fiber-Reinforced Plastics)-Steel Tube

**DOI:** 10.3390/polym14163350

**Published:** 2022-08-17

**Authors:** Qingli Wang, Xiaokang Liu, Kuan Peng

**Affiliations:** 1School of Civil Engineering, University of Science and Technology Liaoning, Anshan 114051, China; 2School of Mechatronic Engineering, Southwest Petroleum University, Chengdu 610500, China

**Keywords:** concrete-filled CFRP steel tube, shearing behavior, experimental study, load carrying capacity, stiffness

## Abstract

Concrete-filled CFRP steel tube (CF-CFRP-ST) structures often suffer from shear loading in practical engineering, such as joints with diagonal braces. To study the shear properties of CF-CFRP-ST, we take the concrete strength and the longitudinal CFRP layers as the main parameters, and static shear tests of overall 9 circular concrete-filled CFRP-steel tube (C-CF-CFRP-ST) and 3 circular concrete-filled steel tube (C-CFST) are carried out. The research is carried out from two aspects: experiment and finite element. The experimental results show that the shear loading-displacement curves of the specimens can be divided into elastic stage, strengthening stage, and softening stage. The increases of the strength of the concrete and the layers of the transverse CFRP can both enhance the shearing load carrying capacity of the specimen. With the increase of concrete strength, there is no obvious change in the shape of the shear stress-shear strain curves of the specimens, and the shear stress and the stiffness of the curve in the elastic stage of the specimen are slightly increased. The shear loading-displacement curves of the specimens are simulated by using finite element software ABAQUS and it is found that the predicted results agree reasonably well with the test results. Then, the whole process of loading and the parameters of the main influencing factors are analyzed. Finally, the calculation equation of CFRP concrete-filled steel tubular shear capacity is established.

## 1. Introduction

Concrete-filled CFRP (carbon fiber reinforcedd polymer) steel tube structure is a structural form in which concrete is filled in CFRP-steel composite tube, and the outer tube and concrete are stressed together [1,2]. The working mechanism of CFRP concrete-filled steel tube is similar to that of traditional concrete-filled steel tube structures and concrete-filled FRP steel tube structures, so that the core concrete is in a three-dimensional stress state by using the constraints of the outer tube, so as to improve the strength of the concrete and improve the mechanical performance of the structure [3,4,5]. In the process of use, concrete-filled CFRP steel tube columns can not only improve the bearing capacity and durability of concrete-filled steel tubular. It can also make full use of the characteristics of CFRP in structural reinforcement; for example, strengthening concrete-filled steel tubular columns after fire can significantly improve their compressive capacity, and CFRP can improve the ductility of hollow steel tubes [6,7].

Liu et al. [8] studied the axial compression bearing capacity of FRP concrete-filled steel tubular. The test parameters include the type and number of FRP, the thickness of steel tubes, and the strength grade of concrete. It is found that the bearing capacity of FRP concrete-filled steel tubular is higher than that of concrete-filled steel tubular. An expression for calculating the bearing capacity of FRP concrete-filled steel tubular is proposed. Li et al. [9] studied the influence of the geometric discontinuity surrounding FRP on the concrete-filled circular steel tube, and calculated the transverse fracture value of FRP using the ideal elastic-plastic bonding model. The finite element calculation results were in good agreement with the test results. The effects of FRP thickness, orthotropic and elastic modulus, as well as bonding yield strength, thickness, and specimen size were investigated. Teng et al. [10] proposed a theoretical model in the form of incremental iteration, which is mainly used to calculate the stress-strain relationship of concrete restrained by circular FRP steel tubes under axial pressure. The results show that the transverse deformation of concrete restrained by FRP steel tube is significantly different from that of concrete restrained by FRP at the initial stage. Zhou et al. [11] analyzed the mechanical properties of CFRP concrete-filled steel tube, and based on the double shear unified strength theory, gave the calculation expression of the axial compression bearing capacity of CFRP concrete-filled steel tube. The calculation results showed that the number of CFRP layers, the tensile strength of CFRP, and the yield strength, wall thickness, and diameter of steel tube were the main factors affecting the bearing capacity of CFRP concrete-filled steel tube. Chen et al. [12] carried out the axial compression test of concrete-filled square steel tubular short columns after CFRP restraint heating. There were 21 columns in total, and the test parameters included the heating temperature of concrete-filled steel tubular short columns and the number of CFRP layers. The test results show that the concrete-filled steel tubular short columns restrained by CFRP can show better mechanical properties. The more layers of CFRP, the higher the initial stiffness and ultimate strength of the specimens. Based on the regression of test data, a simplified formula for the ultimate strength of concrete-filled square steel tubular short columns after CFRP restrained heating was proposed. Mehran Khan and MajidAli [13] proposed the optimization of RC stiffeners using a diagonal approach and validated numerically. The results showed that the behavior of different confined brick masonry structures (CBMS) were compared with those of unconfined brick masonry structure (UBMS). It was concluded that the proposed optimized stiffeners in CBMS meet the required seismic demand for combinations of seismic loadings. Xie CP et al. [14] determined the tensile strength (ft) and fracture toughness (KIC) of reinforced cementitious composites (MHFRCCs) by three-point bending test based on the boundary effect model, and the analysis would be helpful to understand the fracture crack resistance effect of multi-scale hybrid fibers at multi-level within cement matrix.

At present, the research on concrete-filled CFRP steel tube structural members mainly include axial loading, bending moment, compression bending, and other loadings, while there is less available research on shear performance. However, concrete-filled steel tube structural members often suffer from shear loading in practical engineering, such as joints with diagonal braces. Therefore, it is necessary to study their shear properties, and innovatively put forward the finite element model of CF-CFRP-ST bearing shear load. In view of this, nine circular concrete-filled CFRP steel tube specimens and three circular concrete-filled steel tube specimens were designed, and relevant experimental studies were carried out with the concrete strength and the number of transverse CFRP layers as the main parameters to investigate the effects of the above parameters on the shear capacity and stiffness of the specimens. Based on the test, a finite element modeling method was designed to simulate the *V*-*Δ* curve of concrete-filled steel tubular specimens under shear to provide reference for engineering practice.

## 2. Design and Material Properties of Specimens

### 2.1. Design of Specimens

Nine circular concrete-filled CFRP steel tube specimens and three circular concrete-filled steel tubular specimens were tested under shear loading. The main parameters included concrete cube strength *f*_cu_ and transverse CFRP layer number *m*_t_. The length *L* of the specimen was 128 mm, in which the length of the shear span area was *a* = 9 mm, the length of the embedded area at both ends was *b* = 40 mm, and the width of the loading surface was *c* = 30 mm. The size of the specimen is shown in Figure 1.

The outer diameter *D*_s_ of the steel tube was 120 mm, the wall thickness *t*_s_ of the steel tube was 2 mm, and the shear span ratio *λ* = *a*/*D*_s_ = 0.075 < 0.2 [1,15]. At this time, the bending moment of the specimen could be ignored, and it was considered that it only bears the shear loading. Other parameters are shown in Table 1.

*E*_c_ is the elastic modulus of concrete, *ξ*_s_ is the constraint coefficient of steel tube, *ξ*_cf_ is the lateral CFRP constraint coefficient [1], *ξ* is the total constraint coefficient [1], where
*ξ*_s_ = *A*_s_*f*_y_/(*A*_c_*f*_ck_)(1)
*ξ*_cf_ = *A*_cft_*f*_cft_/(*A*_c_*f*_ck_)(2)
*ξ* = *ξ*_s_ + *ξ*_cf_(3)
*f*_cft_ = *E*_cf_*ε*_cftr_(4)
where: *A*_s_ and *f*_y_ are the cross-sectional area and yield strength of the steel tube, respectively; *A*_c_ and *f*_ck_ are the standard values of cross-sectional area and axial compressive strength of concrete, respectively (*f*_ck_ = 0.67*f*_cu_), and *A*_cft_ and *f*_cft_ are the cross-sectional area and tensile strength of transverse CFRP, respectively; and *E*_cf_ and *ε*_cftr_ CFRR are the elastic modulus of carbon fiber cloth and the fracture strain of CFRP, respectively (the measured value in this paper was 5500 με).

All specimens before loading are shown in Figure 2.

### 2.2. Material Properties

The yield strength *f*_y_ = 466 MPa (the shear yield strength *f*_yv_ = 0.58*f*_y_ was taken in the finite element simulation), the ultimate strength *f*_u_ = 610 MPa, the elastic modulus *E*_s_ = 206 GPa, the Poisson’s ratio *v*_s_ = 0.28, and the elongation *δ* = 27% were measured.

The concrete adopted ordinary 42.5 Portland cement (C), grade I fly ash (FA), river sand (S), stone (G) with particle size of 5–15 mm, tap water (W), and water reducer (SP). The mixture proportions are shown in Table 2. The measured *f*_cu_ and *E*_c_ of each group of concrete are listed in Table 1.

Unidirectional carbon fiber cloth was used, with elastic modulus *E*_cf_ = 230 GPa and single-layer thickness *t*_cf_ = 0.111 mm. JGN-C building structure adhesive and JGN-P building structure adhesive produced by Liaoning Academy of construction sciences were used as adhesive and primer, respectively.

## 3. Loading and Measurement

The test device is shown in Figure 3. The estimated bearing capacity of 30% was preloaded first (when estimating the bearing capacity, the transverse CFRP is equivalent to the steel tube according to the principle of equal strength, and then the bearing capacity is estimated by using the formula [16] for the shear bearing capacity of concrete-filled steel tube). The formal loading adopted the graded loading system. In the elastic stage, the load of each level was about 1/10 of the estimated bearing capacity, and the load was held for 2 min. After entering the enhancement stage, loading was carried out slowly and continuously until the specimen was damaged, and unloading after the load dropped to 30% of the peak load.

As shown in Figure 1, strain gauges were pasted on the steel tube and CFRP to measure the strains in three directions at each measuring point, namely, the transverse strain *ε*_t_, the longitudinal strain *ε*_l_ and the 45° direction strain *ε*_45_.

The shear displacement *Δ* of the specimen through 8 displacement meters (Figure 3) was measured,
*Δ* = *Δ*_t_ − *Δ*_s_ − *Δ*_θ_(5)
where: *Δ*_t_ is the total displacement, taking the average value of the displacement measured by the displacement meter (14a) on the diagonal of the jack connecting plate; *Δ*_s_ is the settlement displacement of the embedded support, and the average value of the displacement measured by the displacement meter (14b) at the bottom of the embedded support was taken; *Δ*_θ_. For the displacement caused by the rotation of the support, the displacement caused by the rotation of the support on both sides was taken *Δ*_θ_′ average of
*Δ*_θ_′ = |*Δ*_u_ − *Δ*_d_|(*a* + *b*)/*h*(6)
where: *Δ*_u_ and *Δ*_d_ are the horizontal displacement measured by the displacement meter (14c) at the upper and lower ends of the unilateral embedded support respectively, and H is the height of the embedded support.

## 4. Test Results and Analysis

### 4.1. Test Phenomenon

In the initial stage of loading, the relationship between load and displacement was linear, and the specimen had no obvious deformation. When the load reached about 65% of the peak load, a continuous slight sound could be heard, which was the sound of CFRP cracking, and then it entered the enhancement stage. When the load reached about 90% of the peak loading, the specimen in the loading area was sheared out as a whole, showing that the upper part was concave and the lower part was convex, and the CFRP fracture at 1/2 of the height of the specimen section is shown in Figure 4a. When the peak load was reached, the steel tube at the upper part of the loading area was fractured, and the exposed concrete could be observed. As the lower steel tube was fractured along the edge of the support, the bearing capacity of the specimen began to decline and entered the softening section. With the increase of shear displacement, when the load dropped to about 65% of the peak load, a sharp popping sound could be heard, and inclined cracks appeared in the steel tube at 1/2 of the section height, as shown in Figure 4b. After fracturing the outer tube at the end of the test, it was found that the concrete in the loading area was sheared as a whole, and inclined cracks appeared at the corresponding place to the steel tube cracks, as shown in Figure 4c.

Combined with Figure 4a–c, it can be seen that CFRP, steel tube, and concrete can work together. All specimens after the test are shown in Figure 5.

### 4.2. V-Δ Curves

Figure 6 shows the *V*-*Δ* curve of the specimen. At the initial stage of loading, the shear loading and displacement were linear, the stiffness of the specimen was large, and the concrete and steel tube bore the shear loading together, which belongs to the elastic stage. When the load reached about 65% of the peak load, the steel yielded, the displacement of the specimen increased faster, the shear loading increased slower, and the stiffness of the specimen decreased. When the specimen reached the peak load, the steel tube was sheared and the curve entered the descending section. It could also be seen that with the increase of *f*_cu_, the shear capacity of the specimen increased significantly, and the stiffness in the elastic stage also increased. The reason is that, as the main component of concrete-filled CFRP steel tube, concrete’s strength change has a certain impact on the bearing capacity of the specimen, and the strength change of concrete is positively correlated with the bearing capacity change of the specimen. Increasing *m*_t_ can improve the shear capacity of the specimen, but it has little effect on the stiffness in the curve elastic stage. This is because CFRP can improve the restraint effect on the member to a certain extent, but will not affect the overall elastic modulus of the specimen. Therefore, the increase of the number of CFRP layers under the shear loading can improve the bearing capacity of the member, and the stiffness in the curve elastic stage will not change. The changes of *f*_cu_ and *m*_t_ have little effect on the shape of the curve.

### 4.3. τ-γ Curves

Figure 7 shows the *τ*-*γ* curve of the specimen. It can be seen that with the increase of *f*_cu_, the shear stress of the specimen and the stiffness of the curve elastic stage increased slightly, and the shape of the curve did not change significantly.

## 5. Finite Element Simulation

The stress-strain relationship of steel, concrete, and transverse CFRP, as well as the treatment methods of element selection, mesh generation, and interface model were consistent with those in literature [17,18,19,20,21]. Steel tube, end plate, and concrete were modeled by simplified integrated (C3D8R). Carbon fiber cloth was simulated by four node membrane element M3D4, which has only in-plane stiffness and no bending stiffness. The boundary conditions are shown in Figure 8.

### 5.1. Comparison between Simulation Results and Test Results

#### 5.1.1. Comparison of V-Δ Curve between Simulation Results and Test Results

Figure 9 shows the comparison between the simulation results and test results of the *V*-*Δ* curve of some specimens. It can be seen that the simulation results were in good agreement with the experimental results. The error of between test and finite element is shown in Table 3.

#### 5.1.2. Comparison of Fail Mode between Simulation Results and TEST Results

Figure 10 shows the comparison of test and simulated failure modes of concrete-filled circular CFRP steel tube specimens. It can be seen that the simulation results were in good agreement with the experimental results.

## 6. Analysis of the Whole Process of Stress

### 6.1. Typical V-Δ Curve

Figure 11 shows the typical *V*-*Δ* curve of concrete-filled CFRP steel tube under shear loading. The curve was divided into three stages and seven feature points were selected for analysis. Elastic stage (point *O* ~3): this stage can be divided into three stages, of which point *O* ~1 is the initial stiffness elastic stage. The concrete and steel tube bear the shear load together, and the curve is basically linear. Point 1 corresponds to the yield of steel at the side edge of the loading area. The second elastic stage of stiffness degradation is from point 1 to point 2. 2 points correspond to the concrete cracking near the section in the shear span area. The third elastic stage of stiffness degradation is from point 2 to point 3. The curve is still basically linear, and the stiffness continues to decrease. Point 3 corresponds to the end of the elastic stage of most of the steel in the shear span area. Elastoplastic stage (3–5 points): concrete cracks continue to develop, steel tube and concrete are mainly in the stress state of bidirectional shear, and 4 points correspond to the steel above the middle section of the shear span entering the yield stage. Plastic stage (5–7 points): the shear strain of steel at 1/2 height of the side of the shear span at 5 points is 15,000 με, the transverse CFRP fracture at 6 points, and the ultimate bearing capacity of the specimen at 7 points. The calculation parameters of the specimen are: *L* = 128 mm, *D*_s_ = 120 mm, *t*_s_ = 2.0 mm, *f*_cu_ = 60 MPa, *f*_y_ = 345 MPa, *f*_yv_ = 200 MPa, *ξ*_s_ = 0.349, *ξ*_cf_ = 0.125, *ξ* = 0.474.

### 6.2. Maximum Principal Stress of Concrete

Figure 12 shows the maximum principal stress of concrete. It can be seen that the maximum principal stress was symmetrically distributed along both sides of the loading area during the whole loading process. The concrete in the upper part of the loading area and the lower part of the embedded area was in compression, and the concrete in the shear span area and the upper part of the embedded area were in tension. At point 2, the concrete in the shear span reached the cracking strain, and the maximum principal stress of the concrete reached the maximum value, and then gradually decreased.

### 6.3. Stress of Steel Tube

Figure 13 shows the longitudinal distribution of Mises stress in steel tube. It can be seen that at point 1, under the influence of stress concentration, the steel in the shear span area reached the shear yield strength of 201.5 MPa. At point 4, the steel stress in the shear span area reached 199.4 MPa, and the steel tube began to yield. After point 5, the steel in the shear span area entered the plastic strengthening stage. At point 4, the steel stress in the middle section of the shear span area just reached the shear yield strength of the flat plate area (*f*_yv_ = 266 MPa); At point 7, Mises stress reached the maximum, and then the steel tube fractured.

### 6.4. Stress of CFRP

Figure 14 shows the stress distribution of CFRP. It can be seen that the stress value of transverse CFRP increased continuously at the stage of points 1–6. At point 6, the transverse CFRP stress reached 1261 MPa, basically reaching the ultimate tensile stress of the transverse CFRP of 1265 MPa, and the transverse CFRP began to fracture. The stress value decreased rapidly after point 6.

## 7. Parameter Analysis

Influence parameters to the *V*-*Δ* curve of concrete-filled CFRP steel tube under shear loading included the number of transverse CFRP layers, steel yield strength, concrete strength, and steel ratio. The following is a typical example to analyze the influence of the above parameters on the *V*-*Δ* of concrete-filled CFRP steel tubes. Example conditions: *f*_y_ = 345 MPa, *f*_cu_ = 30 MPa, *α* = 0.07, *D*_s_ = 120 mm, *L* = 128 mm, *m*_t_ = 1.

### Influence of Parameters

Figure 15a–d shows the effects of the number of transverse CFRP layers, the yield strength of steel, the compressive strength of concrete, and the steel ratio on the *V*-*Δ* curve of members under shear. It can be seen that with the increase of *m*_t_, the curve shape and initial stiffness did not change significantly, and the shear capacity of the specimen increased slightly. With the increase of *f*_y_, the shear capacity of the specimen continued to improve, and the initial stiffness of the curve increased slightly, but the shape of the curve did not change significantly. The shear capacity of the specimen and the initial stiffness of the curve increased significantly with the increase of *f*_cu_, but the shape of the curve did not change significantly. With the increase of *α,* the shear capacity of the specimen and the initial stiffness of the curve were improved, but the shape of the curve did not change significantly.

## 8. Bearing Capacity-Related Equation

### 8.1. Calculation Expression

The shear loading corresponding to the maximum shear strain of the members reaching 15,000 με is defined as the shear bearing capacity of concrete-filled CFRP steel tube [22,23,24]. Through the *V*-*Δ* curves, a large number of calculations was carried out for the curve, and the calculation parameters (scope of application): *f*_y_ = 235 MPa–420 MPa, *f*_cu_ = 30 MPa–90 MPa, *α* = 0.05–0.2, *ξ*_s_ = 0.2–4, *ξ*_cf_ = 0–0.6. Finally, the formula of concrete CFRP-filled steel tubular shear capacity calculation coefficient is:*γ*_v_ = 1.0627 − 0.4191ln(*ξ*)(7)

Therefore, the calculation expression of concrete-filled CFRP steel tubular shear capacity is:


(8)
$$V_{u}=\gamma_{v}A_{\mathrm{cfscy}}\tau_{\mathrm{cfscy}}$$



(9)
$$\tau_{\mathrm{cfscy}}=e^{-1.5\xi_{\mathrm{cf}}\left(0.6+0.1\eta +0.313\alpha^{2.33}\right)\xi^{\left(0.134+1.2\xi^{\prime }\right)}f_{\mathrm{cfscy}}}$$


When calculating the shear capacity of concrete-filled steel tubular, the longitudinal CFRP enhancement coefficient *η* = 0.

### 8.2. Expression Validation

Figure 16 shows the comparison between the calculated shear capacity *V*_u_^c^ and the test value *V*_u_^t^ of concrete-filled CFRP steel tubular shear specimens. The average value of *V*_u_^c^/*V*_u_^t^ of the specimen was 0.947, and the mean square deviation was 0.0897. The *V*_u_^c^ and *V*_u_^t^ were in good agreement, and the calculated value was less than the test value, which is partial to safety.

## 9. Discussion

In the present study, nine circular concrete-filled CFRP steel tube specimens and three circular concrete-filled steel tube specimens were designed, and relevant experimental studies were carried out with the concrete strength and the number of transverse CFRP layers as the main parameters to investigate the effects of the above parameters on the shear capacity and stiffness of the specimens. Liang et al. [25] carried out axial compression tests on CFRP-confined recycled aggregate concrete-filled square steel tubular short columns. Wang et al. [19] conducted eccentric pressure tests on circular concrete-filled steel tubular and circular CFRP concrete-filled steel tubular, with a total of 10 specimens, and an expression for calculating the bearing capacity of circular CFRP concrete-filled steel tubular short columns under eccentric compression is proposed. Park et al. [26] conducted compression bending hysteretic tests on existing concrete-filled square steel tubes and concrete-filled square CFRP steel tubes. At this stage, a large number of scholars have studied the various properties of CFRP concrete-filled steel tubular, but have not studied the shear properties. Qian [27] and Xiao [28] studied the shear performance of concrete-filled steel tubular through shear experiment. Shi [29] used the finite element software ABAQUS to simulate the test results and established a reliable finite element model of the shear behavior of concrete-filled steel tubular. Those test results and FE model are basically consistent with those of our three CFST specimens, indicating that our test loading method is reasonable. On this basis, we carried out the performance research of CFRP concrete-filled steel tubular specimens under shear load, and creatively proposed the finite element model of CFRP concrete-filled steel tubular shear specimens. Finally, the shear strength of concrete-filled CFRP steel tube was defined, and the formula for calculating the shear capacity of concrete-filled CFRP steel tube was proposed.

## 10. Conclusions


(1)The shear displacement curve of concrete-filled CFRP steel tube shear specimens can be divided into elastic stage, strengthening stage, and softening stage.(2)The shear displacement curve and failure mode of concrete-filled CFRP steel tube members under shear loading were simulated by ABAQUS, and the simulation results were in good agreement with the experimental results.(3)The typical *V*-*Δ* curve is divided into three stages and seven characteristic points were selected to analyze the stress distribution of the constituent materials in each stage and characteristic points.(4)The results of parameter analysis showed that the increase of steel yield strength or steel ratio can significantly improve the shear capacity, the increase of concrete compressive strength improved the bearing capacity, while the increase of transverse CFRP layers only slightly improved the bearing capacity. The increase of concrete compressive strength and steel content can significantly improve the stiffness of the specimen, and the increase of steel yield strength can improve the stiffness of the specimen.(5)The shear strength of concrete-filled CFRP steel tube was defined, and the formula for calculating the shear capacity of concrete-filled CFRP steel tube was proposed. The results of this formula were in good agreement with the experimental results, and the shear strength can pre-calculate the bearing capacity of concrete-filled CFRP steel tube under shear loading in engineering.


## Figures and Tables

**Figure 1 polymers-14-03350-f001:**
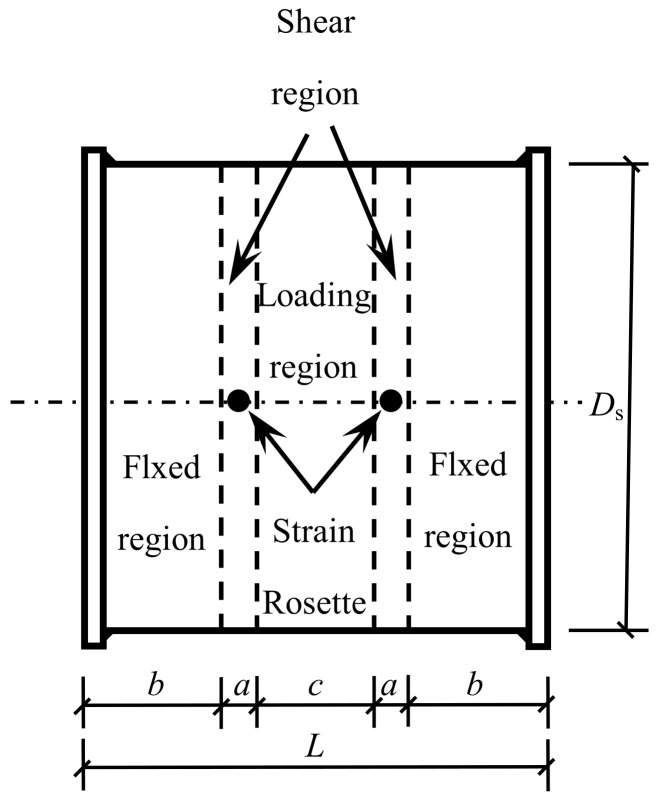
Dimensions of specimens.

**Figure 2 polymers-14-03350-f002:**
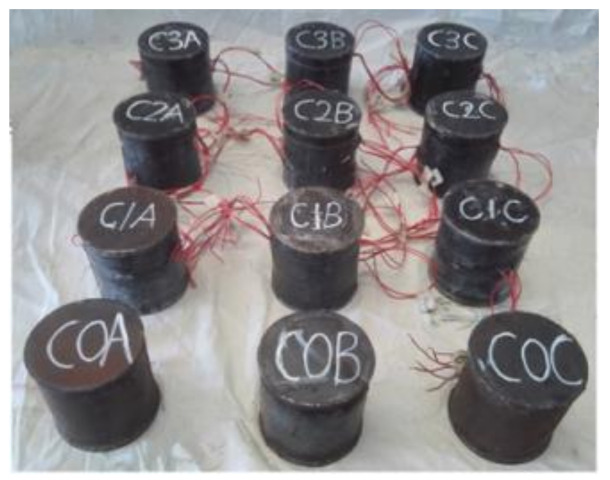
Fabricated CF-CFRP-ST specimens.

**Figure 3 polymers-14-03350-f003:**
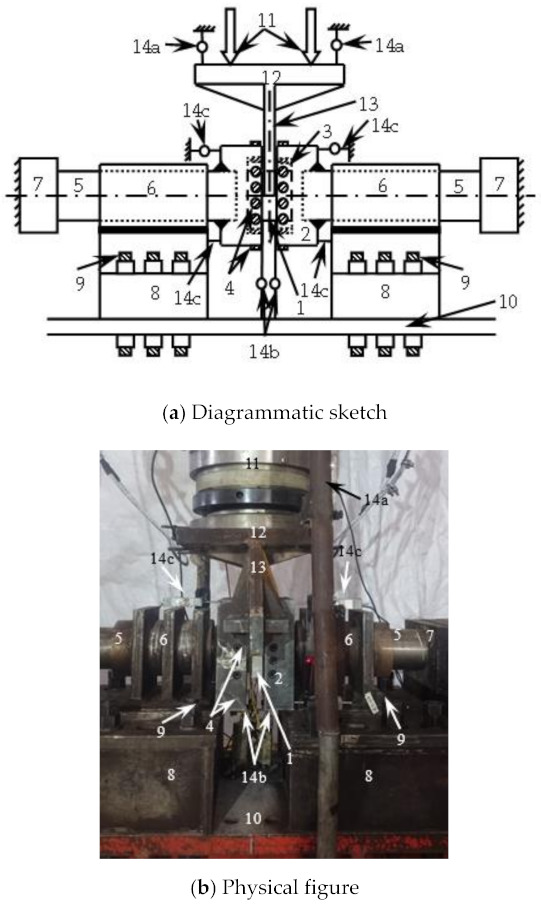
Loading setup. (1. Specimen, 2. embedded support, 3. rigid cushion block, 4. pressure bearing/positioning screw, 5. pressure bearing, 6. pressure bearing sleeve, 7. reaction pier, 8. rigid support, 9. high strength screw, 10. Self-reaction frame, 11. jack, 12. jack connecting plate, 13. rigid fixture, 14. displacement meter).

**Figure 4 polymers-14-03350-f004:**
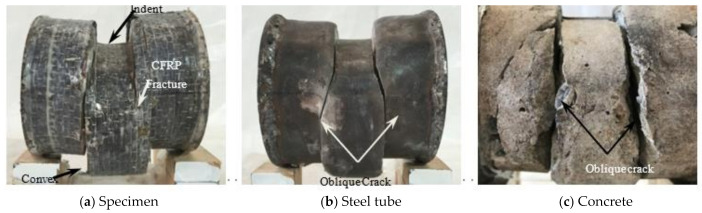
Failure modes.

**Figure 5 polymers-14-03350-f005:**
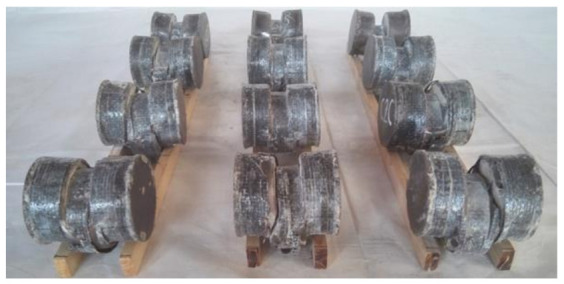
All specimens after testing.

**Figure 6 polymers-14-03350-f006:**
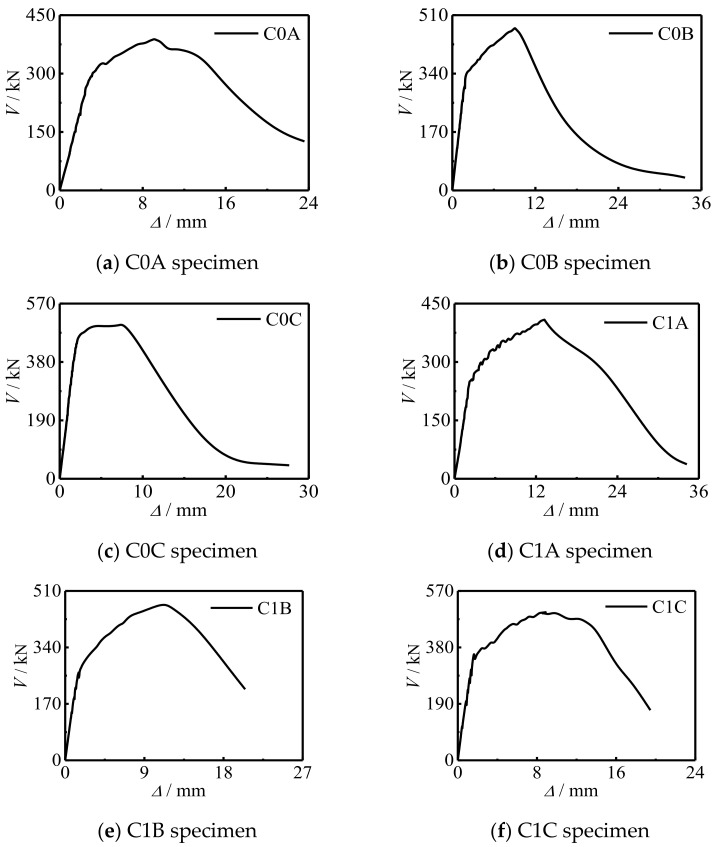
*V*-*Δ* curves of specimens.

**Figure 7 polymers-14-03350-f007:**
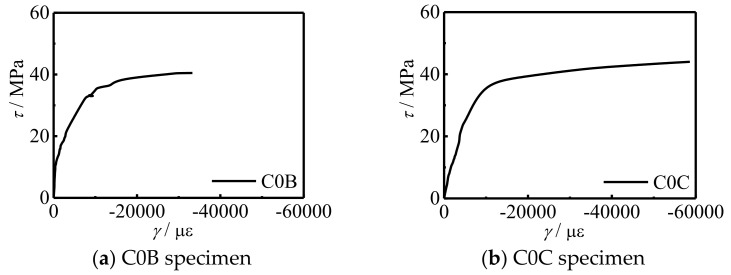
*τ*-*γ* curves of specimens.

**Figure 8 polymers-14-03350-f008:**
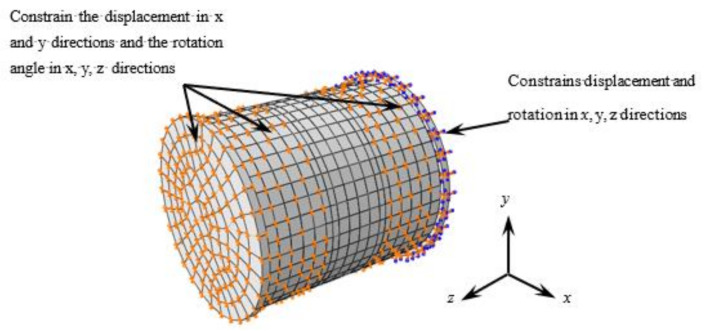
Boundary conditions.

**Figure 9 polymers-14-03350-f009:**
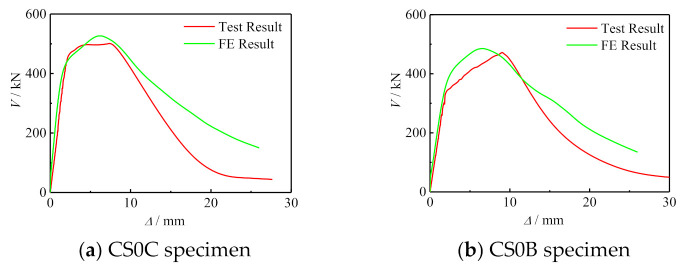
Comparisons between simulated results and tested results of *V*-*Δ* curves of some specimens.

**Figure 10 polymers-14-03350-f010:**
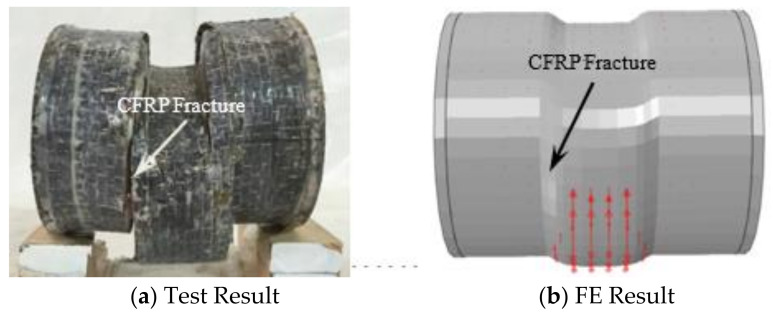
Comparison of test and simulated failure modes of concrete-filled circular CFRP steel tube.

**Figure 11 polymers-14-03350-f011:**
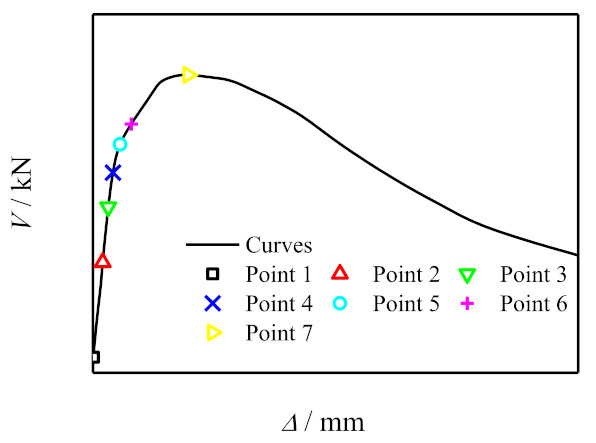
Typical *V*-*Δ* curve of concrete-filled CFRP steel tube under shear loading.

**Figure 12 polymers-14-03350-f012:**
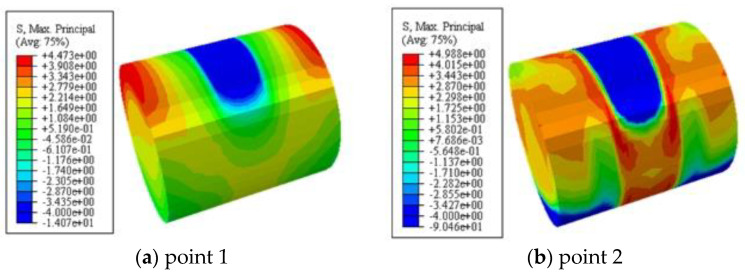
The maximum principal stress of concrete.

**Figure 13 polymers-14-03350-f013:**
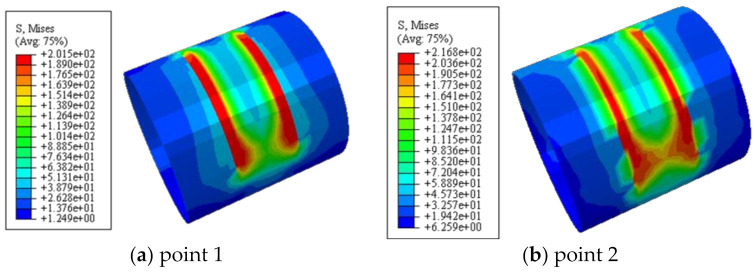
The longitudinal distribution of Mises stress in steel tube.

**Figure 14 polymers-14-03350-f014:**
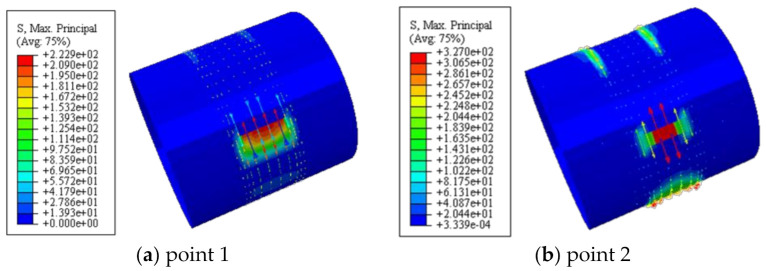
The stress distribution of CFRP.

**Figure 15 polymers-14-03350-f015:**
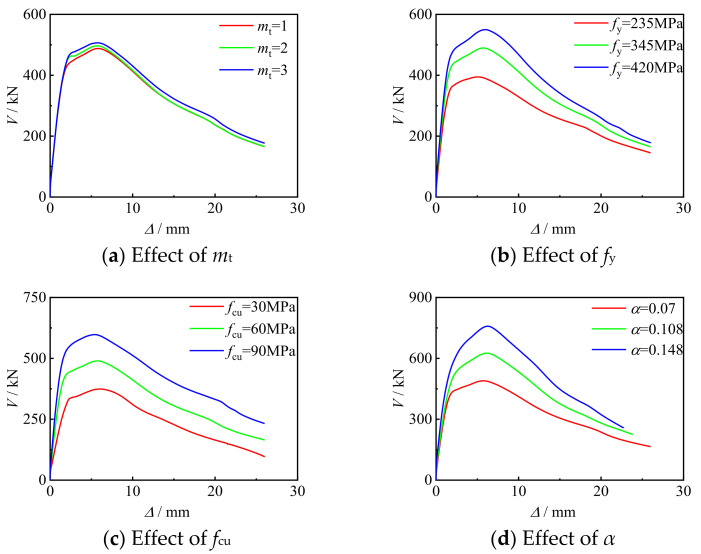
The effects of parameters.

**Figure 16 polymers-14-03350-f016:**
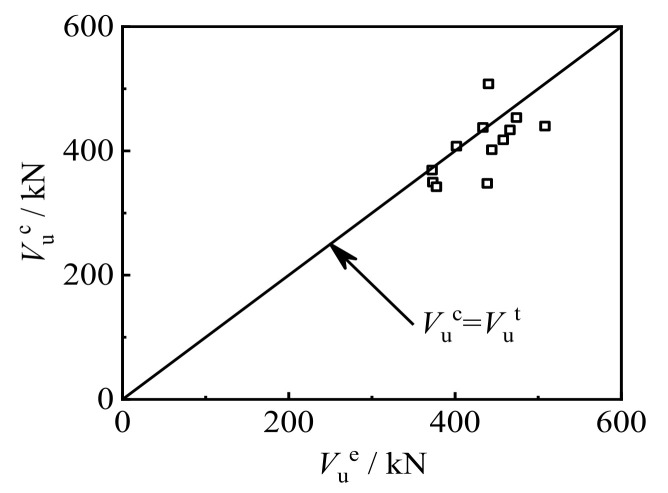
Comparison between *V*_u_^c^ and *V*_u_^t^ of concrete-filled CFRP steel tube shear specimens.

**Table 1 polymers-14-03350-t001:** Several factors of specimens.

No.	*f* _cu_	*E* _c_	*m* _t_	*ξ* _s_	*ξ* _cf_	*ξ*
C0A	35.1	31.5	0	1.38	0	1.38
C1A	35.1	31.5	1	1.38	0.21	1.60
C2A	35.1	31.5	2	1.38	0.42	1.81
C3A	35.1	31.5	3	1.38	0.65	2.02
C0B	46.1	33.2	0	1.05	0	1.05
C1B	46.1	33.2	1	1.05	0.16	1.22
C2B	46.1	33.2	2	1.05	0.32	1.39
C3B	46.1	33.2	3	1.05	0.49	1.99
C0C	54.9	35.5	0	0.88	0	0.88
C1C	54.9	35.5	1	0.88	0.14	1.02
C2C	54.9	35.5	2	0.88	0.27	1.15
C3C	54.9	35.5	3	0.88	0.41	1.29

**Table 2 polymers-14-03350-t002:** Mixture proportions of concrete.

Group	C	FA	S	G	W	SP
A	0.6	0.4	2.5	1.5	0.4	0.01
B	0.6	0.4	2	1.4	0.35	0.01
C	0.74	0.26	1.2	1.5	0.3	0.009

**Table 3 polymers-14-03350-t003:** The error of between test and finite element.

Specimens’ Label	Initial Stiffness of Test (kN/mm)	Initial Stiffness of FE (kN/mm)	Error between Test and FE (%)	Bearing Capacity of Test (kN)	Bearing Capacity of FE (kN)	Error between Test and FE (%)
CS0C	225.17	231.88	2.83	503.15	526.82	4.23
CS0B	164.06	171.77	4.41	472.121	483.246	2.73
CS2B	196.15	201.75	2.7	487.735	508.646	4.22
CS1C	186.78	204.65	8.73	493.05	540.371	9.51
CS2C	222.35	231.06	3.72	512.46	547.557	6.37
CS3C	239.04	250.17	4.99	542.96	552.178	1.93

## Data Availability

No new data were created or analyzed in this study. Data sharing is not applicable to this article.

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
