# Peer review of "Study on Shearing Behavior of Circular Concrete-Filled CFRP (Carbon Fiber-Reinforced Plastics)-Steel Tube"

_polymers, 2022, doi:10.3390/polym14163350_

Round 1
Reviewer 1 Report
Reviewed article concerns study on shearing behavior of circular concrete filled CFRP-steel tube and is write in accordance with generally accepted standards of the scientific works. After careful reading of the submitted text there are some substantive remarks that should be taken into consideration by the Authors to improve reviewed text.
1. The abstract should include information about new methods, results, concepts, and conclusions – in its current form, the abstract needs to be rewritten to include more information on novelty and motivation of the work.
2. The novelty of given approach should be emphasized in introduction.
3. I suggest providing more precise information about used experimental and measurement positions.
4. Presented study widely covers defined scientific problem and with experimental and simulation investigations provides proper background for given conclusions, however deeper scientific consideration of obtained results referred to the basic phenomena in shearing behavior should be given.
5. I suggest also to give wider description of potential use of presented findings in scientific research as well as in industrial practice.
6. In the discussion section should be provide more references to already known results from literature.
7. Conclusions should refer to specific values (results of analysis) as well as basic phenomena that cause described results.
The article has been edited with little care and there are many editorial shortcomings. Some of these are indicated below:
‒ figure cannot be larger than the page (fig. 6, 7) – figure and its caption should be in the same page,
‒ lack of punctuation marks after equitation (equitation is part of a sentence),
‒ language correction is needed,
‒ consequently, all values should be writing with space before its unit (with very few exceptions).
Author Response
| 1. The abstract should include information about new methods, results, concepts, and conclusions – in its current form, the abstract needs to be rewritten to include more information on novelty and motivation of the work. |
The authors thank the reviewer to present this suggestion. Authors rewrite abstract, and add information on novelty and motivation of the work. Please see the highlight text (in yellow colour) in abstract.
|
|
|
2. The novelty of given approach should be emphasized in introduction. |
The authors agree with the reviewer’s comments in this point. The novelty of approach is emphasized in introduction. Please see the highlight text (in yellow colour) in introduction. |
|
|
3. I suggest providing more precise information about used experimental and measurement positions. |
Some precise information is added in Fig.1 and Fig.3. |
|
| 4. Presented study widely covers defined scientific problem and with experimental and simulation investigations provides proper background for given conclusions, however deeper scientific consideration of obtained results referred to the basic phenomena in shearing behavior should be given. |
Thanks for your careful review. We have analyzed the influence of various material performance changes on specimens under shear load in the parameter analysis part and basic phenomena is explained it in detail. |
|
| 5. I suggest also to give wider description of potential use of presented findings in scientific research as well as in industrial practice. |
The shear strength of concrete-filled CFRP steel tube is defined, and the formula for calculating the shear capacity of concrete-filled CFRP steel tube is proposed. The shear strength can pre-calculate the bearing capacity of concrete-filled CFRP steel tube under shear loading in engineering. Please see the highlight text (in yellow colour) in conclusion. |
|
| 6. In the discussion section should be provide more references to already known results from literature. |
The authors agree with the reviewer’s comments in this point. A large number of references are supplemented in discussion. |
|
| 7. Conclusions should refer to specific values (results of analysis) as well as basic phenomena that cause described results. |
Conclusions are revised according to the comment and basic phenomena are added. Please see the highlight text (in yellow colour) in conclusion. |
|
| The article has been edited with little care and there are many editorial shortcomings. Some of these are indicated below: ‒ figure cannot be larger than the page (fig. 6, 7) – figure and its caption should be in the same page, ‒ lack of punctuation marks after equitation (equitation is part of a sentence), ‒ language correction is needed, ‒ consequently, all values should be writing with space before its unit (with very few exceptions). |
Thanks for your careful review. Figures are edited according to the journal. Punctuation marks after equitation are added. Problem of language are checked and revised, and all values is writing with space before its unit. Please see the highlight text (in yellow colour) in 6 and 7.1 section. |

Reviewer 2 Report
1. Title “ CFRP” should written in first appear in detail.
2. Keywords : Circular concrete filled CFRP-steel tub – is very long
3. Do the dimensions which taken according to the national or international standard ?
4. Figures labels should be detailed well
5. Discussions related to figure.6 should show the difference between each sample, the same with figure.7 as it looks to summarize the results only.
6. Element types should be mentioned in FEM.
7. A clear comparison including table to summary your experimental and FEA results and compare with literature is recommended.
8. Results should be supported with refences and deep discussions
9. Most of the reference are not up to date.
Author Response
|
1. Title “ CFRP” should written in first appear in detail. |
Title “ CFRP” is written in first appear in detail. Please see the highlight text (in yellow colour) in title. |
|
2. Keywords : Circular concrete filled CFRP-steel tub – is very long |
The authors agree with the reviewer’s comments in this point. Authors revise this keyword.
|
|
3. Do the dimensions which taken according to the national or international standard ? |
Thanks for your careful review. The design of specimens refers some references and national standard. Those references are added. Please see the highlight text (in green colour) in 1.1 section. |
|
4. Figures labels should be detailed well |
Some Figures labels are detailed. |
|
5. Discussions related to figure.6 should show the difference between each sample, the same with figure.7 as it looks to summarize the results only. |
Figure.6 or figure.7 show V-D curve and t-g curve of all specimens. Those figures are just a demonstration, and the performance of each specimen under different influencing factors is specifically described in the parameter analysis section. |
|
6. Element types should be mentioned in FEM. |
The authors agree with the reviewer’s comments in this point. Element types are mentioned. Steel tube, end plate and concrete are modeled by simplified integrated (C3D8R). Carbon fiber cloth is simulated by four node membrane element M3D4, which has only in-plane stiffness and no bending stiffness. Please see the highlight text (in yellow colour) in 4 section. |
|
7. A clear comparison including table to summary your experimental and FEA results and compare with literature is recommended. |
The error of between test and finite element is shown in Table 3. Please see the highlight text (in yellow colour) in Section 4.1. |
|
8. Results should be supported with refences and deep discussions |
The authors agree with the reviewer’s comments in this point. Some references are added in manuscript to support conclusions. |
|
9. Most of the reference are not up to date. |
Thanks for your careful review. We have added many updated references in the text and Introduction. |
Reviewer 3 Report
The paper is very interesting and good. The authors had made a significant contribution by studying the shearing behavior of circular concrete filled CFRP steel tube. Following are my minor suggestion.
1. Please modify the title that cover all aspects.
2. Add more details about types of finite element models and other fracture details in the introduction. Please refer to the https://doi.org/10.1016/j.jobe.2020.101689 ; https://doi.org/10.1016/j.compositesb.2021.109219
3. Too long paragraphs in introduction. Please reduce them and make it specific.
4. Please write clearly the objective and significance of your work in the last paragraph of introduction.
5. Please improve all the Figure quality especially Figure 3.
6. A comparison with previous study is necessary.
7. Add a discussion section before conclusion regarding practical implementation of current study.
8. Please make bullet point in conclusion.
9. What are your future recommendation.
10. Moderate English changes required.
11. Merge Figure 6 and try to add more in one Figure instead of reporting them single.
12. Similarly make Figure 7 concise and shorter.
13. Make text clear in Figure 12-14.
14. References are too less and too old. Add at least 30 from recent years.
Author Response
|
1. Please modify the title that cover all aspects. |
Thanks for your suggestion. Title is modified and the full name of CFRP is added. |
|
2. Add more details about types of finite element models and other fracture details in the introduction. Please refer to the https://doi.org/10.1016/j.jobe.2020.101689 ; https://doi.org/10.1016/j.compositesb.2021.109219 |
Those references are added in section introduction. Please see the highlight text (in green colour) in Section introduction. |
|
3. Too long paragraphs in introduction. Please reduce them and make it specific. |
Thanks for your careful review. Some paragraphs are reduced in Introduction. |
|
4. Please write clearly the objective and significance of your work in the last paragraph of introduction. |
We revised objective and significance of our work in the last paragraph of introduction. |
|
5. Please improve all the Figure quality especially Figure 3. |
The authors agree with the reviewer’s comment in this point. Figure 3 is revised, and relevant annotations are added. Please see the highlight text (in yellow colour) in Section 2. |
|
6. A comparison with previous study is necessary. |
Due to the lack of previous research on the shear performance of CFRP concrete-filled steel tubular, which is also the first time we have studied it, we designed three ordinary concrete-filled steel tubular specimens as comparison to study, and compared the shear performance of CFRP concrete-filled steel tubular and concrete-filled steel tubular. |
|
7. Add a discussion section before conclusion regarding practical implementation of current study. |
Thanks for your comment. Discussion section is added in paper before conclusion. |
|
8. Please make bullet point in conclusion. |
The authors agree with the reviewer’s comment in this point. We revised the conclusion and divided the conclusion into five bullet points |
|
9. What are your future recommendation. |
In the future, we will consider to study the performance of CFRP concrete filled steel tubular under other loads and conduct some finite element simulation for it. |
|
10. Moderate English changes required. |
Thanks for your careful review. Problem of language are checked and revised. |
|
11. Merge Figure 6 and try to add more in one Figure instead of reporting them single. |
In the initial version of this article, we merged Figure 6 and Figure 7, but after merging, a large number of curves coincided, and the situation is not saw clearly, so we separated the curves in order to show them more clearly. For the influence of various factors on the curves, we described them in detail in the parameter analysis part. For Figure 6 and Figure 7, this part can be put into the supplementary materials of the article in the future, which does not occupy the page. If the reviewer really needs to merge those figures, we can merge it in the next revision. |
|
12. Similarly make Figure 7 concise and shorter.
|
|
|
13. Make text clear in Figure 12-14. |
We enlarged figures 12 to 14 to make the text inside clearer. |
|
14. References are too less and too old. Add at least 30 from recent years. |
The authors agree with the reviewer’s comment in this point. We have supplemented a large number of references and replaced some old references. |
Round 2
Reviewer 1 Report
The authors have revised their manuscript in accordance to the reviewers' comments.
Reviewer 2 Report
Authors replay to the reviewer comments